# Can you hear me? Playback experiment highlights detection range differences between commonly used PAM devices: C-POD, F-POD and SoundTrap

**Nicole R. E. Todd**[1,2]*, **Ailbhe S. Kavanagh**[3], **Mark J. Jessopp**[1,2], **Willem Verboom**[4], **Emer Rogan**[2]

**1** MaREI Centre, Environmental Research Institute, University College Cork, Cork, Ireland, **2** School of Biological, Earth & Environmental Sciences (BEES), University College Cork, Cork, Ireland, **3** Fisheries Ecosystems Advisory Services (FEAS), Marine Institute, Oranmore, Galway, Ireland, **4** JunoBioacoustics, Winkel, The Netherlands

* nicole.todd@ucc.ie

## Abstract

Passive acoustic monitoring (PAM) is a valuable tool for monitoring acoustically active small cetaceans such as the harbour porpoise (*Phocoena phocoena*), with a range of devices commonly used across studies. However, to ensure comparability of findings, there is a need to compare the ability of devices to detect acoustic signals. Using a playback approach, we determined the detection probability and effective detection radius/area (EDR/EDA) for co-deployed C-POD (Cetacean POrpoise Detectors), F-POD (Full waveform capture POD) and SoundTrap acoustic monitoring devices. We conducted playbacks of harbour porpoise recordings across two transects at a range of distances from moored devices, while accounting for a range of variables likely to influence the detection probability of playbacks. Distance from the devices influenced the detection probability across all devices, and a significant difference between transects was also found for the C-POD, possibly due to different ambient noise conditions. The maximum detection distance of the playbacks for the SoundTrap and the F-POD was between 400 - 500m, and EDR was estimated at 297m (EDA 0.276 km$^2$) and 241m (EDA 0.181 km$^2$), respectively. The maximum detection distance for the C-POD was lower, at 300 - 400m, and an EDR of 220m (EDA 0.153 km$^2$). A lower EDR was calculated for harbour porpoise buzzes compared to clicks across devices, due to lower source level of buzzes, suggesting that time spent foraging may be underestimated in PAM studies. The results highlight how detection ranges may differ across commonly used PAM devices, affecting comparability of detection rates across studies. EDR/EDA is an important prerequisite for PAM-derived density and abundance estimates. As such, understanding how devices differ is essential for comparing studies and appropriate planning of long-term acoustic monitoring projects, particularly where estimates of abundance are a key goal.

**Data availability statement:** All processed data and R code files to reproduce the results given in this paper are available in the following repository https://github.com/Ntodd95/playbacks

**Funding:** This publication has emanated from research conducted with the financial support of Taighde Éireann – Research Ireland under Grant number [GOIPG/2019/2173]. Open access funding is provided by IReL. The funders had no role in study design, data collection and analysis, decision to publish, or preparation of the manuscript.

**Competing interests:** The authors have declared that no competing interests exist.

## Introduction

For sound-producing animals, passive acoustic monitoring (PAM) is an efficient, non-invasive technique to sample populations and make inferences about species distribution and occurrence over space and time (e.g., [1]). Passive acoustic methods are well suited for monitoring cetacean species that are difficult to monitor visually, such as deep diving species (e.g., beaked whales) [2], or smaller cetaceans with cryptic surfacing behaviour, including several species of porpoise [3]. The use of PAM-derived data to estimate animal density and abundance is becoming more widely used and has been reviewed for both terrestrial and marine settings [4–5]. The application of such studies has been successful in the marine environment, in particular for cetaceans, where acoustic methods have been used to estimate the density of *inter alia* fin whales (*Balaenoptera physalus*) [6], deep diving Blainville's (*Mesoplodon densirostris*) [7], goose-beaked (*Ziphius cavirostris*) beaked whales [8–9] and harbour porpoise [10].

Various PAM technologies have been used for studying cetaceans, including towed or drifting hydrophone arrays [11–13], fibre optic cables [14], ocean gliders [15] and bottom-mounted static devices (e.g., [16]). Bottom-mounted, static PAM devices can be loosely categorised as either those that record full bandwidth raw acoustic data (e.g., SoundTrap, Ocean Instruments; Autonomous Multichannel Acoustic Recorders (AMARs), JASCO Applied Sciences; and HydroMoth, Open Acoustic Devices) or as click detectors with onboard detection algorithms that store acoustic information about the detected sounds (e.g., Cetacean POrpoise Detectors (C-PODs) and its successor, the Full waveform capture PODS (F-PODs), Chelonia Ltd.). Click detectors record information such as date and time, peak frequency, and amplitude, but they do not retain raw acoustic files for processing. Static PAM devices provide high-resolution data over long monitoring periods at discrete locations and have been widely used to monitor small coastal cetaceans such as the harbour porpoise (e.g., [17–21]). To better interpret the resultant long-term PAM datasets, and ensure appropriate coverage of a survey area, it is important to consider the detection ranges of the devices used for the species of interest.

The detection range of a static PAM device is often referred to as the effective detection radius (EDR). This is a key prerequisite for density and abundance estimation using PAM [5]. The EDR is defined as the distance at which the number of missed detections near the detector is equal to the number of recorded detections at greater distances, within a maximum truncation distance [9]. Subsequently, EDR can be converted to an effective detection area (EDA), representing the spatial area monitored, that can be used for calculations of density and abundance. To obtain accurate detection distances, and estimate EDR/EDA for static PAM devices, studies have used simultaneous visual and acoustic surveys [22–23], as well as playbacks of artificial cetacean clicks or recordings of echolocation behaviour [10,24]. For monitoring harbour porpoises, EDR has been estimated for T-PODs (the predecessor of the C-POD [22]), as well as more recently for F-POD and SoundTrap [25] while EDA has been estimated for C-PODs [10,24]. However, to the best of our knowledge, no studies have compared the detection range of these commonly used PAM devices within the same soundscape setting. Many variables may influence the propagation of sound through water, including water temperature, depth, ambient noise, salinity, and seabed substrate (through transmission loss/absorption by the sediment or through obstruction from physical objects) [1,26].

Comparative studies have shown that SoundTraps can detect 50% more harbour porpoise detections when deployed alongside a C-POD [27–28]. While the primary reason for this difference remains unclear, the C-POD's zero-crossing detection algorithm has been suggested to be more affected by ambient noise than the detection and classification algorithms used with SoundTraps. Furthermore, co-deployed C-POD and F-PODs showed that the F-POD consistently logged significantly more harbour porpoise detections and detected nuanced temporal patterns in foraging behaviour that the C-POD failed to detect [29]. Detection differences between PODs also

introduce uncertainty in the comparability between devices and lead to implications for long-term monitoring studies. This poses the questions of (i) how comparable static PAM devices are in terms of the detection range, and (ii) what factors influence the detection rates across devices. This study aims to address this knowledge gap by (i) estimating the detection ranges (as EDR & EDA) of three commonly used static PAM devices within the same soundscape, and (ii) assessing the effect harbour porpoise call type, including echolocation clicks and buzzes, on detection ranges.

## Materials and methods

### Deployment site and conditions

The deployment site is a relatively sheltered site on the southwest coast of Sherkin Island (SW Ireland, 51°27′40.7" N, 9°26′24.7" W), situated within Roaringwater Bay and Islands Special Area of Conservation (SAC). This SAC is designated for and supports harbour porpoise throughout the year [30]. Three static PAM devices were deployed, attached to an anchored mooring line in approximately 18m water depth, in an area with a predominantly sandy seafloor. A C-POD and an F-POD were co-deployed side by side in a custom-built acetal plastic frame secured to a mooring line at a depth of approximately 13m, with a SoundTrap600HF (Ocean Instruments, New Zealand) deployed immediately above this on the same line. The SoundTrap was configured on high gain, with a high pass filter (predefined in SoundTrap Audio options) and a sampling frequency of 384kHz.

The playback experiment was conducted on 02/04/2023 with a water temperature of 10°C at the deployment site. Playbacks of harbour porpoise acoustic signals were conducted from a rigid inflatable boat, drifting with engines off to avoid engine noise in a sea state of Beaufort 2 (wind speed of 4 - 6kts, small wavelets with no whitecaps). The engine was briefly on at the start of the playback experiment but was turned off for the remainder of the experiment to avoid interference. This was not expected to impact the results and was not investigated further. Following consultation with the National Parks and Wildlife Service it was established that no permits or additional permissions were required to conduct the research, however the following measures were implemented to reduce potential disturbance to wild cetaceans. Before commencing playbacks, a 30-minute visual survey of the area was conducted for the presence of any cetacean species, and playbacks were only carried out if no animals were sighted within a visible radius of approximately 5 km from the vessel and PAM mooring. Visual observations were also conducted during the transmission of the playbacks to ensure that no cetacean species knowingly entered the area during the playback experiment.

### Playback recordings

Recorded acoustic signals from captive harbour porpoises were used to investigate the detection ranges of C-POD, F-POD and SoundTrap in the field. Echolocation clicks were recorded from two harbour porpoises at Harderwijk Marine Mammal Park, Netherlands, housed in an oval concrete pool (8.6 x 6.3m, water depth 1.3m) as part of a systematic study on harbour porpoise acoustic signals [31–33]. The selected recordings were chosen based on their representativeness of harbour porpoise echolocation signals and hence deemed suitable for assessing the detection range of the passive acoustic monitoring (PAM) devices through a playback experiment. Recordings were made using a hydrophone (Bruel and Kjaer 8101) suspended in the centre of the pool at mid-water depth, connected to an amplifier (TPD) and a high-speed tape recorder (Racal Store 4D) [31]. The recorded signals were digitised at a sampling frequency of 352,800 Hz at 16-bit. Recorded signals contained a mix of porpoise echolocation behaviour, including echolocation clicks and buzzes. Artificial signals were not used because producing such signals was outside the scope of the available resources for this study. Summary details on each of the recordings used during the playback experiment are outlined in Table 1. General observation clicks, i.e., regular spatial-orientation echolocation

**Table 1. Summary details of the recorded harbour porpoise acoustic signals. General observation refers to a free-swimming porpoise scanning the environment; Object investigation refers to targeted high repetition signals used to identify target objects.**

|  | Duration (sec) | Recorded echolocation | Activity | Por-poise | Peak fre-quency | Associated literature for more information | Calculated source lev-els (re 1 μPa @ 1 m) |
|---|---|---|---|---|---|---|---|
| **Rec1** | 0.55 | 10 clicks + buzz | General observation and object investigation | adult | 126 kHz | [34] | 125 dB |
| **Rec2** | 2.8 | HF buzz | Object investigation | young | 140 kHz | [35] | 125 dB |
| **Rec3** | 2.3 | 2x buzz | Object investigation | adult | 116 kHz | [36] | 120 dB |
| **Rec4** | 2.25 | 3x buzz + clicks | General observation and object investigation | adult | 116 kHz | [37] | 123 dB |
| **Rec5** | 0.72 | 6 double clicks | General observation | adult | 132 kHz | [38] | 132 dB |

clicks, were recorded from a free-swimming porpoise scanning its environment. Due to this scanning behaviour, the majority of the echolocation clicks were off-axis, i.e., porpoise clicks emitted away from the centre of the echolocation beam, rather than emitted at maximum power of a directional beam. Object investigation buzzes, i.e., high repetition signals used to identify target objects (and often used in highly discriminatory tasks such as foraging or social behaviour), were recorded from a stationary porpoise at the centre beam (i.e., on-axis, targeted echolocation).

Most of the clicks recorded contained both high-frequency and low-frequency components [38]. Harbour porpoise adapt their source level according to their environment [39] and circumstances (i.e., when performing certain tasks). Maximum porpoise source levels are expected to be around 175 dBrms or around 190 dBpp (peak to peak) re 1 μPa/1m in open-water conditions. As the recording of the acoustic signals took place in a small pool, it was not possible to measure the emitted levels due to the acoustic reflections within the pool. In addition, click properties and pulse shape may have differed from recordings conducted in free-field, or open-water conditions.

## Playback experiment

Playbacks of harbour porpoise acoustic signals were transmitted at 100m increments from the PAM mooring, starting at 100m, and ranging up to a maximum of 600m from the moored static PAM devices, resulting in six playback stations per transect. The position of the transects was randomly chosen in the field to ensure diverse coverage of the area. Two transects were conducted at different bearings to the moored devices, and each transect had two replicates (Fig 1). Transect one took place on a bearing of 295° North at the farthest playback station, and the heading was maintained for the duration of the transect, although some slight variation was recorded due to the drift of the boat. Transect 2 was conducted at a bearing of 8.5° North.

The distance of each playback station to the moored static PAM devices was determined using the boat's GPS, calculating a linear distance from the marked device position and GPS waypoint of the boat. GPS waypoints were recorded for each playback station to ensure that the exact locations could be revisited for the replication of the transect. Measured distances were recorded as "true distance" to account for any minor deviations. In most cases, these were within 5m of the intended station marks; however, for Transect 1, the 100m station was recorded at 128.75m and the 200m station at 225.97m. As all stations were subject to minor positional shifts caused by boat drift during playback transmission, hence this was not presumed to alter the results.

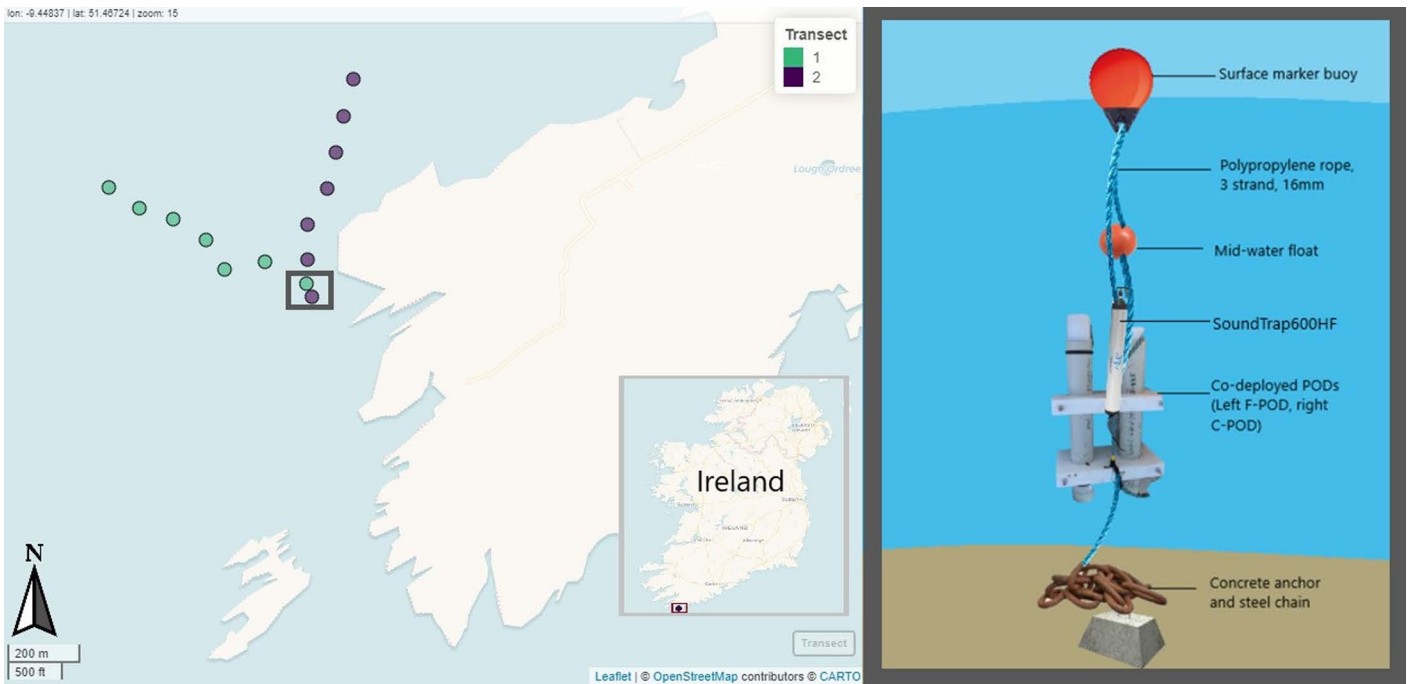

**Fig 1. Left: Map showing the locations of the two transect lines conducted for the playback experiment, with playback stations at increments of 100m to 600m from the PAM devices indicated by individual points.** Grey box indicates the position of the moored PAM devices. Note that the moored devices were in slightly different latitudinal/longitudinal positions between the morning and afternoon due to slight movement due to currents and tide, the distance between the two points was 35m. Inset map shows the location of the experimental site in the context of Ireland. Right: Diagram of mooring set-up for the PAM devices.

Replicates of transect lines 1 and 2 were completed following the same trajectory, directly after the completion of the respective transect to increase the comparability of replicates. Playbacks along transect 1 were conducted in the morning, whereas those conducted along transect 2 were carried out in the afternoon. GPS waypoints for the moored PAM devices were recorded at the start of both transect 1 and transect 2 to account for any change in positioning throughout the day due to tidal state and currents.

Recorded harbour porpoise acoustic signals were transmitted using a computer running PAMGUARD connected to a National Instruments Ltd 6361 usb-box and amplified by a VP2000 amplifier. Playbacks were conducted using the Sound Output module in PAMGUARD, configured using the Sound Acquisition module to play each selected recording on repeat for one minute. Playback signals were transmitted through an omnidirectional piezo-electric transducer (TC4033: Teledyne Reson A/S, Denmark, with a transmitting sensitivity of -137 dB re 1 µPa/V for 130 kHz signal) which was suspended from the boat at a depth of approximately 2m below the water surface.

Each playback station consisted of five distinct recordings of harbour porpoise signals (n = 5), each played sequentially at two amplification levels (30 dB and 50 dB gain). The sequence of playbacks followed the pattern: Rec 1 (30 dB), Rec 1 (50 dB), Rec 2 (30 dB), Rec 2 (50 dB), Rec 3 (30 dB), and so forth. There was no time gap between individual play-backs; recordings were played in direct succession according to this sequence. The gain settings (30 dB and 50 dB) were selected to represent low and high amplification levels appropriate for the playback equipment and to evaluate potential differences in detection

performance at varying source levels. Playback details, including recording number, gain setting, and playback timing, were recorded in the field and then later transcribed into the playback2.2 csv dataset (see repository), which provides a complete log of all playbacks. Each transect was replicated twice, leading to a total of 20 playbacks per station per transect. With two transects conducted, this resulted in a total of 40 playbacks per 100m mark sampled.

In addition to the playback experiment, a SoundTrap placed 1m from the transducer, at the same depth, was used to monitor the playback output Sound Pressure Level (SPL). Source levels are expressed in dB relevant to a reference unit and can be defined as SPL at 1m from the source [26,40]. SoundTrap data were exported via the SoundTrap Host software, and.wav file was imported to Audacity where SPL was extracted using the *Measure SPL* plugin. The SoundTrap was factory-calibrated with a receiving sensitivity of -176.3 dB re 1 μPa/V. Sensitivity is provided for both high and low gain settings and represents the SPL for a normalised (±1.0) wav file with a full-scale signal, i.e., 2.0 units peak to peak.

Unlike traditional hydrophone systems, SoundTraps integrate the recorder and hydrophone, resulting in a fixed relationship between sound pressure and the resultant wav file data, eliminating the need for voltage calculations. Audacity reports SPL in units of dB re full scale. To scale the SoundTrap data, we followed methods outlined in the SoundTrap user guide [41], where the SPL value (dB re 1 μPa) equals the Audacity SPL reading, plus end-to-end calibration (−176.3 dB in this case) minus 3dB. SPL measurements were taken from a test setup conducted at close range prior to the field experiment, with separate measurements recorded for playbacks at both 30 dB and 50 dB gain settings. A single SPL measurement was taken across the full spectrum for each recording to account for potential frequency variation, particularly where multiple call types were present. Source levels were calculated between 120 and 132 dB re 1 μPa @ 1 m (Table 1), with some variation due to differing peak frequencies.

## Data analysis

Data from the deployed C-POD and F-POD were exported using FPOD.exe Version 1.1 [42]. To assess which playbacks were detected by each static PAM device, the raw click files (CP1 & FP1) were initially inspected. Field data forms were consulted to identify the timestamps of each playback, and it was noted whether a valid detection (i.e., at least one click recorded within the playback period within the appropriate frequency range) was identified. This resulted in binary data as detected (1) or not detected (0). CP1 and FP1 files were further processed by Chelonia's KERNO train classifier into CP3/FP3 files to identify if any of the playbacks were identified as a click train, either in the correct species class of NBHF (Narrowband high frequency, i.e., harbour porpoise in this case) or as a click train but with unclassified species class. As the classifier requires a minimum of six clicks to identify a train, we analysed raw detections to assess whether the playbacks were detected by the devices, independent of classification. However, instances where the classifier identified a click train during playback were also noted.

SoundTrap data were exported via the SoundTrap Host software, and.wav files were imported to Audacity for visual inspection of the spectrograms during the playback period (see Figs in S1 File for inspection criteria). As with the data analysed from the C-POD and F-POD, a playback was noted as detected (1) or not detected (0) for the periods corresponding to each playback.

## Detection function model fitting

Based on distance sampling theory, the probability of detection can be described as a function of distance to the PAM device [43]. The detection function was modelled for each static PAM device using a binary Generalised Additive Model (GAM), with "detected" (1) or "not detected" (0) as the binary response variable following similar methods outlined in previous research [10,24]. Distance of the playback from the devices was included as an explanatory variable along with a suite of additional variables that may influence the detection of a playback. These included transect number, source level, water depth (range 18 - 40m), and tide (as a continuous variable between 0-3 representative of the rule of twelfths, based on a smooth rate of tidal flow reaching a maximum velocity halfway between low and high tide). All playbacks from both gain settings were included in the analyses, with source level incorporated as an explanatory variable in the models to account for variations between recordings and across gain settings. All numerical variables were modelled as thin-plate regression splines, limited to a maximum of 5 degrees of freedom to avoid overly complicated models. Transect number was included in the model as a factor. Stepwise model selection was performed where non-significant interactions were dropped from the model (starting with the least significant) and model validation was repeated. Models were compared using AIC to choose the best and final model.

The GAM was implemented using the *gam* function in the *mgcv* Package [44], with a binomial error structure and logit link function. Variance and 95% confidence intervals (CIs) were calculated using a nonparametric bootstrap. This approach was used to estimate the detection function g(x) [43], i.e., the probability of detecting a porpoise playback given it is at distance x from the device. The fitted detection functions were then used to estimate the Effective Detection Radius (EDR) for each static acoustic device [22] (1).

$$\hat{P} = \frac{2}{w^2} \int_0^w x \, \hat{g}(x) \, dx \tag{1}$$

$\hat{P}$ is computed as the average probability of detecting a porpoise playback within distance $W$ of the detector. Based on standard distance sampling applications, this assumes uniform detections over the space surveyed by the detectors (i.e., within some horizontal distance $W$ (truncation distance), at which detection probability is assumed to be 0). Here we used a truncation distance of 500m based on previous studies [10], and then integrated out distance *x*. The EDR ($\hat{p}$) was calculated from the estimated probability (P) using the following equation (2):

$$\hat{p} = \sqrt{\hat{P} W^2} \tag{2}$$

A similar methodology was also applied [10,24] to compute the Effective Detection Area (EDA) of static acoustic monitoring devices as a prerequisite for density estimation. EDA is defined as a horizontal circle around the logger within which many detections are missed as are detected outside the circle during a 1-second period (based on distance sampling frameworks, e.g., [43]). This is calculated in two dimensions rather than three dimensions as animal density is calculated per unit area and not volume. Variation in EDA due to differences in depth can be to some extent captured by including depth as a covariate in analyses [10]. Although no density estimation was carried out in the current study, to calculate EDA (*V*), EDR was converted to EDA (3), for each of the devices to investigate comparability with previous studies (division by 1,000,000 is to produce a result in km$^2$):

$$\nu = \frac{\pi^* \hat{p}^2}{1,000,000} \tag{3}$$

As described above, EDA is defined in two dimensions although PAM is typically omnidirectional, represented by a sphere in three dimensions. Actual detection capabilities in terms of EDR/EDA will be influenced by the Directivity Index (DI) of the hydrophone used and PAM detections are only recorded when the porpoise transmitting beam (approx. +/− 16 degrees around the primary axis) is directed towards the hydrophone. Consequently, for comparison we also calculated the detection range of the devices in a spherical representation, defined as Effective Detection Volume (EDV) (4). EDV represents the theoretical volume in which PAM can detect porpoise playbacks, assuming open water and ignoring the influence of bottom or surface (e.g., reflections or sediment absorption), also assuming an omnidirectional sensor (DI = 0 dB).

$$EDV = \frac{\frac{4}{3} * \pi * \hat{p}^3}{1,000,000,000} \tag{4}$$

These analyses were repeated for each static PAM device. They were also used to investigate the effect that porpoise call type (i.e., spatial orientation echolocation clicks hereafter referred to as clicks, or buzzes) may have on the EDR of playbacks. As the recorded signals used for the playbacks represented a variety of porpoise echolocation behaviour, a subset of the data was selected for specific analysis. Recording 3 was chosen to represent a "buzz only" playback, as it consisted exclusively of buzzes, while recording 5 was chosen to represent a "click only" playback, containing solely clicks. The distinct echolocation behaviour in each of these recordings made them suitable as representatives of the two call types, as the other recordings contained a mix of buzzes and clicks and were not suitable for this analysis. In the GAM's used to investigate the effect of call type, distance and recording number were included as the explanatory variables, while all other model components remained consistent with the previous analysis.

## Results

There were no cetacean sightings throughout the duration of the playback experiment. Across the two transects, 240 playbacks of harbour porpoise signals were conducted. Of these playbacks, 100 were detected by the SoundTrap (42%), 90 were detected by the F-POD (38%), and 63 were detected by the C-POD (27%) (Table 2). Detection of the playbacks classified per recording are also presented in Table 2. Of the playbacks detected by the C-POD, 34 were recognised as click trains by the Chelonia KERNO classifier with 12 classified as NBHF trains. The KERNO-F classifier for the F-POD identified 18 of the playback sequences as click trains, while only 2 were classified as NBHF origin. No playbacks were detected on any of the static acoustic monitoring devices beyond the 500m stations (Table 2). The SoundTrap and the F-POD both recorded maximum detection distances below the 500m range, while the C-POD had maximum detection of the playbacks within 300m of the recording devices.

GAMs explained 68.4-81.1% of the deviance across the static PAM devices (Table 3). There was a significant increase in the detection of playbacks on all devices with decreasing distance (Table 3). For the SoundTrap and C-POD, in addition to distance, there was an increase in detections of playbacks with an increased source level. The detection of playbacks by the C-POD was also significantly influenced by transect, with a greater number of detected playbacks from transect 1. Tidal flow was not found to be significant in any of the models and was omitted during the model selection process.

**Table 2. Number of detected playbacks at each playback station for each device. A total of 40 playbacks were transmitted at each station across all recordings combined, including both gain settings, with 8 playbacks per individual recording. Results are presented for all recordings combined and further classified by individual recording.**

| Distance of playback station to PAM devices (m) | SoundTrap | | | | | | F-POD | | | | | | C-POD | | | | | |
|---|---|---|---|---|---|---|---|---|---|---|---|---|---|---|---|---|---|---|
| | All Rec | Rec1 | Rec2 | Rec3 | Rec4 | Rec5 | All Rec | Rec1 | Rec2 | Rec3 | Rec4 | Rec5 | All Rec | Rec1 | Rec2 | Rec3 | Rec4 | Rec5 |
| 600 | 0 | 0 | 0 | 0 | 0 | 0 | 0 | 0 | 0 | 0 | 0 | 0 | 0 | 0 | 0 | 0 | 0 | 0 |
| 500 | 0 | 0 | 0 | 0 | 0 | 0 | 0 | 0 | 0 | 0 | 0 | 0 | 0 | 0 | 0 | 0 | 0 | 0 |
| 400 | 4 | 2 | 2 | 0 | 0 | 0 | 1 | 0 | 0 | 0 | 0 | 1 | 0 | 0 | 0 | 0 | 0 | 0 |
| 300 | 25 | 8 | 4 | 3 | 4 | 6 | 13 | 5 | 3 | 2 | 1 | 2 | 1 | 0 | 0 | 0 | 0 | 1 |
| 200 | 32 | 8 | 7 | 3 | 6 | 8 | 36 | 8 | 6 | 6 | 8 | 8 | 22 | 8 | 3 | 3 | 2 | 6 |
| 100 | 39 | 8 | 8 | 8 | 8 | 7 | 40 | 8 | 8 | 8 | 8 | 8 | 40 | 8 | 8 | 8 | 8 | 8 |
| Total | 100 | 24 | 19 | 14 | 18 | 21 | 90 | 21 | 17 | 16 | 17 | 18 | 63 | 17 | 11 | 11 | 10 | 14 |

**Table 3. Final model summaries including deviance explained (Dev. Expl.) for GAMs investigating the factors affecting the detection of harbour porpoise playbacks for all PAM devices. Significant interactions are indicated by p-value < 0.05, and bold text. Dashes indicate that the factor was excluded in the model selection process. AIC values for model selection shown in supplement material (see S2 File).**

| | SoundTrap model (Dev. Expl. −68.4%) | | F-POD model (Dev. Expl. −75%) | | C-POD model (Dev. Expl. −81.1%) | |
|---|---|---|---|---|---|---|
| | p-value | Effect size | p-value | Effect size | p-value | Effect size |
| Distance | **<0.001** | 38.48 | **<0.001** | 30.13 | **<0.001** | 17.38 |
| Depth | 0.058 | 3.58 | 0.104 | 7.82 | – | – |
| Source level | **<0.001** | 3.85 | – | – | **0.003** | 3.00 |
| Transect | 0.101 | 1.64 | – | – | **0.009** | 2.62 |
| Tidal flow | – | – | – | – | – | – |

The probability of detection of the playback of harbour porpoise signals was estimated for each PAM device, along with 95% bootstrap confidence intervals (Fig 2). Based on this detection probability, estimated EDR was predicted for all static acoustic monitoring devices (Table 4). The calculated EDR for detecting the playbacks of harbour porpoise acoustic signals under the same conditions was the greatest for the SoundTrap, estimated at 297m (95% CI: 239-363) (Fig 3). The EDR for the F-POD was 241m (95% CI: 211 −282) and the EDR for the C-POD was the lowest at 220m (95% CI: 202 −242) (Fig 3). The fitted detection function model was also used to predict an EDA per device (Table 4). An EDA of 0.276 km² was calculated for the SoundTrap, 0.181 km² for the F-POD, and 0.153 km² for the C-POD (Fig 3). The Effective Detection Volume (EDV) which represents the detection of harbour porpoise playbacks in three dimensions is as follows: SoundTrap - 0.11 km³, F-POD - 0.059 km³, and C-POD - 0.045 km³. This highlights that the omnidirectional detection range of the F-POD is 54% that of the SoundTrap, while the C-POD's range is 41% comparatively (Table 4).

A subset of the data was used to investigate the effect of playbacks of different call types, i.e., click only, or buzz-only. These models showed a significant difference between detection of clicks and buzzes for the SoundTrap, but not for either the F-POD or C-POD. Lower EDR was estimated for the playback of acoustic signals consisting solely of echolocation buzzes, however, it can be noted that CIs are large in some cases reducing the reliability of estimates (Table 4). The EDR of 319m for buzz playback was 73m less than the EDR of 264m for click playback for the SoundTrap. A difference of 31-33m in EDR was found for the C-POD and F-POD despite the relationship not being significant in the models.

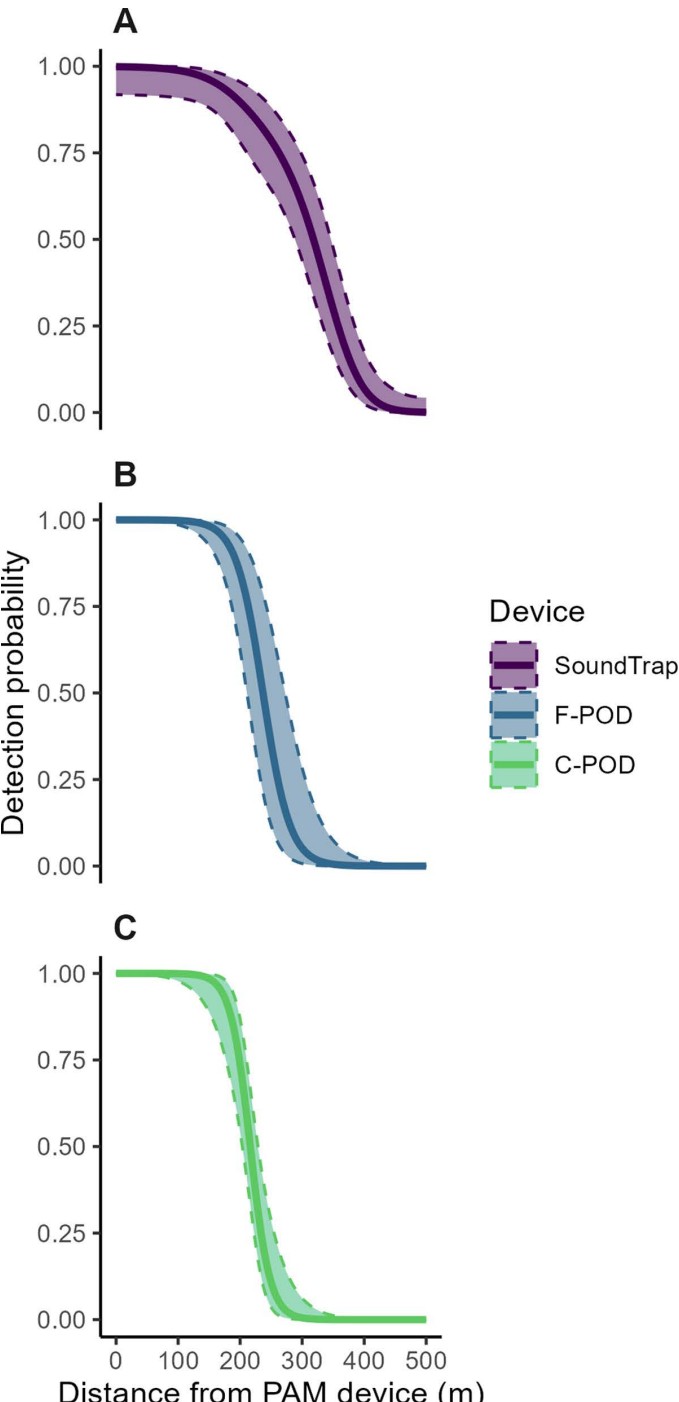

**Fig 2. Estimated probability of detection of all harbour porpoise playbacks (solid smooth lines) and 95% bootstrap confidence intervals (smooth dashed lines and shaded area) for A) SoundTrap, B) F-POD, and C) C-POD.**

**Table 4. Calculated Effective Detection Radii (EDR) and Effective Detection Area for the PAM devices, and EDR for the playback of distinct call types.**

| SAM Device | All Data Analysis | | | Distinct call types analysis | |
|---|---|---|---|---|---|
| | EDR | EDA | EDV | Call type | EDR |
| SoundTrap | 297m (95% CI: 239-363). | 0.276 km² | 0.11 km³ | Click only | 319m (95% CI: 254 - 475) |
| | | | | Buzz only | 246m (95% CI: 168 - 406) |
| F-POD | 241m (95% CI: 211-282) | 0.181 km² | 0.059 km³ | Click only | 296m (95% CI: 248 - 350) |
| | | | | Buzz only | 263m (95% CI: 216 - 317) |
| C-POD | 220m (95% CI: 202-242) | 0.153 km² | 0.045 km³ | Click only | 235m (95% CI: 195 - 291) |
| | | | | Buzz only | 204m (95% CI: 161 - 253) |

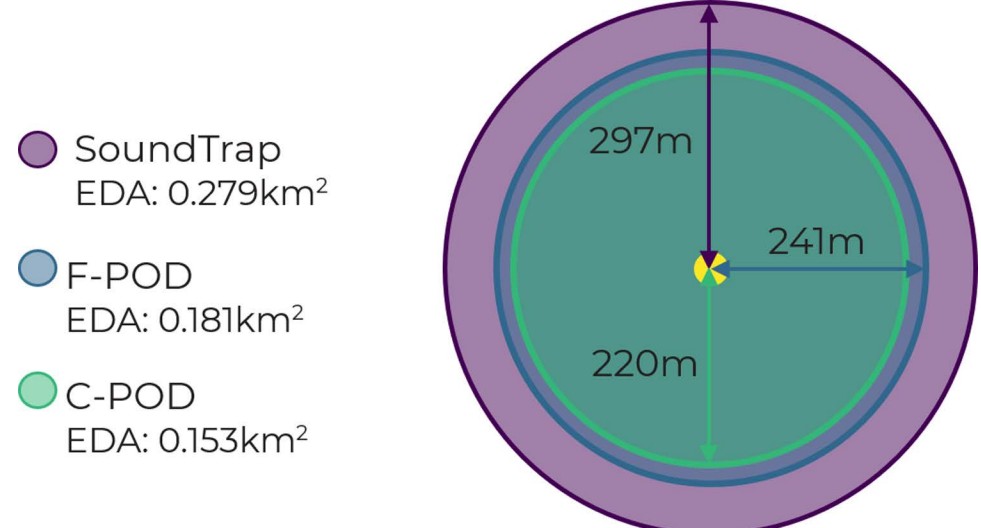

**Fig 3. Detection ranges of SoundTrap, F-POD and C-POD shown as Effective Detection Radius (EDR), and Effective Detection Area (EDA) based on the detection probability of harbour porpoise playbacks.** Date of experiment: 02/04/2023. Estimated probability of detection of the playback of all harbour porpoise vocalisations (solid smooth lines) and 95% bootstrap confidence intervals (smooth dashed lines) for A) SoundTrap, B) F-POD, and C) C-POD.

## Discussion

Long-term passive acoustic studies are often integral to monitoring cetacean occurrence, behaviour, abundance, and habitat preference and subsequent changes across these factors. Choosing the appropriate device when establishing a monitoring programme is key, as comparative studies have shown that static acoustic devices may vary in detection rates even when deployed simultaneously [27–29]. Differing detection capabilities between devices can make it difficult to compare studies, or relative abundance between areas using different devices. Such discrepancies have been shown between C-PODs and full bandwidth SoundTraps [27–28], and between C-PODs and its F-POD successor [29]. There is a clear need to quantify the differences in detection rates as a function of detection ranges. This study addressed this fundamental knowledge gap by comparing the detection ranges of harbour porpoise playbacks for three commonly used PAM devices: C-POD, F-POD and SoundTrap. The maximum detection ranges of the SoundTrap and the F-POD were found to be similar (400 - 500m), and both greater than that of the C-POD. The SoundTrap detected the greatest number of

playbacks overall, highlighting the greater detection capacity of continuous recording hydrophones. Results also showed that there were differences in the detection ranges based on call type, with the detection range of buzzes consistently less than that of clicks across all PAM devices, likely due to reduced source levels. The differences in detection range between devices emphasize the need for careful consideration when choosing PAM devices for monitoring. This is particularly important when monitoring protected areas or offshore developments. It is vital that species or populations are sampled as consistently as possible over time to detect genuine changes in occurrence or density and ensure important areas are afforded appropriate protection.

The estimated EDA for the SoundTrap (0.279 km$^2$) was the greatest of all three static PAM devices and almost double the estimated EDA of the C-POD (0.153 km$^2$). The detection range of the F-POD (0.181 km$^2$) was also found to be greater than the C-POD EDA for this site, although less than the mean C-POD EDA (0.219 km$^2$) reported from the Baltic region [10]. However, it should be noted that considerable variation was found in the EDA across different months and monitoring stations throughout the Baltic study (range: 0.034 km$^2$- 0.742 km$^2$), reflecting different environmental conditions with different soundscapes across the region. Our results show that the detection probability of porpoise playbacks under the same environmental conditions and ambient soundscape, varied between devices. No playbacks were detected on any of the moored devices beyond 500m, in line with the maximum detection range of wild porpoise echolocation assumed by Sveegaard *et al.* [11]. However, it should be emphasized that the EDR depends on the source level of the playback device, and source levels of the playbacks were lower than maximum source levels, so the present study may have shown EDR above 500m if higher SLs had been used.

Despite the SoundTrap and F-POD having the same maximum detection distances, the estimated EDR of the SoundTrap was considerably greater, 297m versus 241m respectively, reflecting the greater number of playbacks at the 300m and 400m stations detected by the SoundTrap. It was also estimated that the omnidirectional detection range of the F-POD, in terms of EDV, was 54% of the SoundTrap, while C-POD was estimated at 41% (Table 4), highlighting large differences between the SoundTrap and the click detectors. Greater consistency in detection of playbacks on the SoundTrap may reflect differences in device sensitivities and critical bandwidth, or the SoundTrap's enhanced ability to detect echolocation clicks from broadband recordings compared to click detectors such as C-POD/F-POD. The click detectors in-built detection algorithms likely also influence how the playbacks, and as such in-situ detections are recorded by the devices compared to full spectrum hydrophones such as the SoundTrap. For the C-POD, our EDR of 220m is comparable to previous research by Nuuttila *et al.* [24] who reported a mean EDR of 188m for harbour porpoise, but it should be noted that our estimate was for one device, and EDR has been shown to vary considerably between C-POD devices [24]. A recent study derived an estimated detection range for SoundTrap and F-POD at a maximum of 105m through tracking wild echolocating porpoises with drones [25]. This range is notably less than the detection ranges derived in the current study, highlighting the high variability in wild porpoise detection ranges due to environmental conditions, device differences, and natural source level variations.

Furthermore, it is wise to assume that variability between individual PAM devices also exists. However, while some variation between the same device type could be attributed to differences in sensitivity, these should have a relatively minor effect if the devices are correctly and regularly calibrated [45]. Additionally, while the performance of classifiers was not considered in great depth in the current study, the results indicated that the F-POD detected more playbacks overall but less were classified as click trains or NBHF, suggesting that the classifiers may have limitations in detecting weaker signals of short duration in playback scenarios,

which could affect performance in long-term monitoring applications. The greater detection ranges found in this study for the SoundTrap and F-POD in comparison to the C-POD could, at least in part, explain previous research showing increased detection rates for harbour porpoise from the SoundTrap [27–28], and the F-POD [29]. While advances in technology resulting in increased detection rates and improved detection ranges is positive, it is important to consider what consequences this may have for comparison with historical data, given that these may underestimate occurrence and abundance compared to newer devices. Interpreting data from newer devices may also be confounded, as declines in species abundance could be potentially masked by the increased detection rates of fewer individuals. This highlights that comparing data derived from multiple devices is a very nuanced matter that should be given careful consideration.

Increasing the source level of the playback increased the detections on both the SoundTrap and the C-POD, which is consistent with previous studies [10,24]. It's important to note that the source level varied between recordings and was also influenced by the gain settings used, with greater detections observed at higher gain settings. Surprisingly source level was not a significant variable for the F-POD suggesting detection algorithms may be less dependent on source level, potentially focusing on signal features such as frequency modulation. For the C-POD only, there was a significant difference in the detection of the playbacks between the two transects (p = 0.009). As the effect of transect was not shown for the other devices, it is unlikely that time of day (with transect two conducted in the afternoon as opposed to the morning), or orientation of device in water (which would influence co-deployed devices) are key drivers of this difference. One plausible explanation for the decreased detection probability for transect two is the potential influence of environmental noise, which may be greater for transect two due to its position closer to land and in slightly shallower water. C-PODs have been reported to be highly sensitive to environmental noise levels [46] within the 20-160 kHz noise band. This may mask porpoise detections and can cause C-PODs to 'max out', limiting further porpoise detections. Studies on F-PODs have shown that environmental noise has less influence on detection rates [29], and do not 'max out' under high ambient noise conditions. It is therefore important to consider the environment and ambient noise levels when choosing deployment sites for PAM equipment.

The EDR of buzzes was lower than that of clicks for all devices. However, the difference in EDR for the SoundTrap was twice as large as the difference observed for the other devices. Other studies on cetaceans have shown that buzzes can typically have lower sound output levels than clicks, with differences of up to 20dB reported for sperm whales (*Physeter macrocephalus*) [47], Blainville's beaked whales [48], and 10dB for harbour porpoise [49]. In the current study, calculated source levels for the "buzz only" playback were 8dB lower than those of the "click only". Therefore, the reduced EDR across the devices is primarily driven by the lower source levels; although, it is also possible that factors such as peak frequency and the number of clicks may have impacted the detection of the two call types. While the click characteristics of wild harbour porpoise may differ from those used in the current experiment, the primary call types were investigated, within the same soundscape, highlighting detection variability that can impact monitoring. Buzzes are a good indication of porpoise foraging behaviour [50–51] and can help identify critical foraging habitat. If buzzes are consistently detected at reduced distances from static PAM devices than clicks, due to reduced source levels, foraging behaviour derived from PAM devices is potentially underestimated due to the reduced search area. It is important to note that wild harbour porpoise echolocation behaviour is highly directional [52–53], and for clicks, this is partly compensated by head-scanning behaviour to ensonify a greater area [50]. However, for buzzes, echolocation is targeted towards an object, e.g., prey, in a close-range focused beam. Given an omnidirectional transducer was used to

conduct this playback experiment, detection ranges of wild porpoise (for all call types) are potentially less than those estimated across the acoustic devices due to increased directionality of their echolocation [54].

In conclusion, this study found significant differences in detection ranges of harbour porpoise playbacks across three commonly used static PAM devices. We provide EDR and EDA for these devices under the same ambient soundscape. As C-PODs reach the end of their operational lives and studies adopt or integrate F-PODs or SoundTraps into acoustic monitoring programmes, increased detection ranges are an undoubted benefit, but consideration needs to be given to the comparability with older devices in time-series analysis. These findings highlight the importance of considering device-specific detection ranges for accurately estimating density and abundance, as well as for designing effective monitoring strategies, particularly for species of concern for conservation and management. If feasible, cetacean monitoring projects using static PAM devices should determine the EDR/EDA to aid comparability of detection rates and to normalise density and abundance estimates across sites and studies.

## Supporting information

**S1 File. Criteria for identifying positive detections of playbacks based on spectrograms.**
(DOCX)

**S2 File. AIC values used in the model selection process.** Model parameters and AIC values for model iterations used in model selection. Stepwise model selection was performed where non-significant interactions were dropped from the model (starting with the least significant) and model validation was repeated. Models were compared using AIC to choose the best and final model. Chosen model highlighted in bold red for each device.
(DOCX)

## Acknowledgments

We would like to thank Nick Tregenza for the loan of the F- POD that supported this project, and the technical support provided by the Chelonia team. We would also like to thank Sam Cox for providing the SoundTrap for this experiment. We are grateful to Ollie Boisseau, Mats Amundin, Hanna Nuuttila, Jamie MacAulay, Magnus Wahlberg and Luke Rendell for insightful discussions. We are grateful to Michael Collins skipper of the Kestrel, and for various field assistants for facilitating the fieldwork in Roaringwater Bay.

## Author contributions

**Conceptualization:** Nicole R. E. Todd, Ailbhe S. Kavanagh, Mark J. Jessopp, Emer Rogan.

**Data curation:** Nicole R. E. Todd, Willem Verboom.

**Formal analysis:** Nicole R. E. Todd.

**Funding acquisition:** Nicole R. E. Todd, Ailbhe S. Kavanagh, Mark J. Jessopp, Emer Rogan.

**Investigation:** Nicole R. E. Todd.

**Methodology:** Nicole R. E. Todd.

**Project administration:** Nicole R. E. Todd.

**Supervision:** Ailbhe S. Kavanagh, Mark J. Jessopp, Emer Rogan.

**Validation:** Nicole R. E. Todd.

**Visualization:** Nicole R. E. Todd.

**Writing – original draft:** Nicole R. E. Todd.

**Writing – review & editing:** Nicole R. E. Todd, Ailbhe S. Kavanagh, Mark J. Jessopp, Willem Verboom, Emer Rogan.

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
