## [Decision Letter · Decision Letter 0]

3 Dec 2024

PONE-D-24-43959Can you hear me? Playback experiment highlights porpoise detection range differences between commonly used PAM devices: C-POD, F-POD, and SoundTrap.PLOS ONE

Dear Dr. Todd,

Thank you for submitting your manuscript to PLOS ONE. After careful consideration, we feel that it has merit but does not fully meet PLOS ONE’s publication criteria as it currently stands. Therefore, we invite you to submit a revised version of the manuscript that addresses the points raised during the review process.

We look forward to receiving your revised manuscript.

Kind regards,

Vitor Hugo Rodrigues Paiva, Ph.D.

Academic Editor

PLOS ONE

https://journals.plos.org/plosone/s/file?id=ba62/PLOSOne_formatting_sample_title_authors_affiliations.pdf "

2. In your Methods section, please provide additional information regarding the permits you obtained for the work. Please ensure you have included the full name of the authority that approved the field site access and, if no permits were required, a brief statement explaining why

“This work was funded by the Irish Research Council Government of Ireland Postgraduate Scholarship Scheme (Project ID: GOIPG/2019/2173).”

“This work was funded by the Irish Research Council Government of Ireland Postgraduate Scholarship Scheme (Project ID: GOIPG/2019/2173). We would like to thank Nick Tregenza for the loan of the F- POD that supported this project and the technical support provided by the Chelonia team. All those who provided discussions during the conception of this study including Ollie Boisseau, Mats Amundin, Hanna Nuuttila, Jamie MacAulay, Magnus Wahlberg and Luke Rendell. We would also like to thank Sam Cox for providing the SoundTrap for this experiment. We are grateful to Michael Collins skipper of the Kestrel, and for various field assistants for facilitating the fieldwork in Roaringwater Bay. Open access funding provided by IReL.”

“This work was funded by the Irish Research Council Government of Ireland Postgraduate Scholarship Scheme (Project ID: GOIPG/2019/2173).”

Reviewers' comments:

Reviewer's Responses to Questions

**Comments to the Author**

1. Is the manuscript technically sound, and do the data support the conclusions?

Reviewer #1: Yes

Reviewer #2: Yes

2. Has the statistical analysis been performed appropriately and rigorously? 

Reviewer #1: Yes

Reviewer #2: Yes

3. Have the authors made all data underlying the findings in their manuscript fully available?

Reviewer #1: Yes

Reviewer #2: Yes

4. Is the manuscript presented in an intelligible fashion and written in standard English?

Reviewer #1: Yes

Reviewer #2: Yes

5. Review Comments to the Author

Reviewer #1: This is a useful article especially to researchers conducting long-term monitoring using different devices, with good insights on EDR/EDA implications. As I mentioned, the duration of the recording itself (or recording type) could be a factor in the reception of the signal by the PAM device, and therefore should be investigated if it is a relevant factor. It is well-written and the structure is clear and concise, but I suggest improving the figures and adding the figure from the supplementary to the article (see my comment in the pdf). I would also like to see more details on the results of model selection.

Reviewer #2: Dear Authors

Thank you for conducting the research and preparing this manuscript comparing the detection performance of commonly used acoustic recorders. I found the manuscript to be well written, concise, and interesting. I think this manuscript will be of great interest to the audience of this journal. I don't have a lot comments except for the few in the marked up document.

6. PLOS authors have the option to publish the peer review history of their article (what does this mean? ). If published, this will include your full peer review and any attached files.

**Do you want your identity to be public for this peer review?** For information about this choice, including consent withdrawal, please see our Privacy Policy .

Reviewer #1: No

Reviewer #2: No

---

## [Author Response · Author response to Decision Letter 1]

18 Dec 2024

Dear Editor, Many thanks for taking the time to provide editorial and reviewer comments on our manuscript “Can you hear me? Playback experiment highlights porpoise detection range differences between commonly used PAM devices: C-POD, F-POD, and SoundTrap” and inviting a revised version. We would like to thank the reviewers for their positive and constructive comments and have outlined below how we have addressed them, with responses below each comment. We hope you agree that the manuscript is much improved and now meets the standard for publication in PLOS ONE.

Reviewer 1

Reviewer #1: This is a useful article especially to researchers conducting long-term monitoring using different devices, with good insights on EDR/EDA implications. As I mentioned, the duration of the recording itself (or recording type) could be a factor in the reception of the signal by the PAM device, and therefore should be investigated if it is a relevant factor. It is well-written, and the structure is clear and concise, but I suggest improving the figures and adding the figure from the supplementary to the article (see my comment in the pdf). I would also like to see more details on the results of model selection.

Response: We thank the reviewer for their positive feedback and helpful comments on our manuscript. We have taken the feedback on board, improving the figures and adding additional information on the models (particularly related to AIC). In relation to the duration of the recordings, all playbacks were transmitted as a 1-minute playback loop. Because the calls were standardised across all stations, any differences between devices are explored by including variables such as source level and type of vocalisations in further analysis. We have also provided a breakdown of detections by call type in Table 2 that shows all calls were detected with almost 100% success by all devices, at least at short distances, so recording duration in this case is unlikely to be a factor in terms of differences in detectability between devices. Please see specific responses to the detailed comments below.

Detailed comments:

Line 25: Determined the

Response: Amended as suggested

Line 45-47: insert reference

Response: Reference added

Line 48-49: in parenthesis

Response: Amended as suggested

Line 90: suggest replacing “and” with “while

Response: Amended as suggested

Line 98: “as much as” these words can be removed

Response: Amended as suggested

Line 112: I'm missing the information on when the deployment and playbacks were conducted, which is important for literature comparison. Water temperature, and therefore season, is also an important factor in the detection of sound (sound speed is affected by temperature).

Response: We agree the context of season and water temperature is important, so a sentence has been added to note when the experiment took place and the water temperature- “The playback experiment was conducted on 02/04/2023 with a water temperature of 10°C at the deployment site.”

Line 117: would be nice to add a figure of the sensor setup next to the map

Response: Figure of senor setup has been added next to the map

Line 183: “Inset map”

Response: Amended as suggested

Line 219: word “files” doubled

Response: Duplicate word deleted

Line 241: Duration of the playback could also affect detection. Perhaps it should be added as a variable?

Response: Duration of playback was standardised across all experiments, with each playback lasting a total of one minute at each playback station. Therefore this was not considered as a variable as it did not vary between playbacks.

Line 270: delete “by”

Response: Amended as suggested

Line 291: Since for EDV, it assumes an omnidirectional sensor, this sentence only applies to calculations of EDA/EDR?

Response: For clarity, we have moved this sentence and added direct reference to EDR/EDA.

Line 300: I can see now that recording number is included, which in a way includes duration (but also vocalization type and other factors). But maybe also interesting to see duration as a separate explanatory variable..

Response: All playbacks were transmitted for a total of one minute duration, there should be no difference between the recordings in terms of duration so we did not consider this as an explanatory variable.

Line 300: And 3 or 5 in parentheses are not clear to me

Response: 3 or 5 was deleted as was potentially misleading and it is described in the previous sentence which recordings are used in the subset

Line 315: classify each detection by recording type in this table. is the reception of the device influenced by the recording type (and not necessarily the vocalisation type as you mentioned)?

Response: As suggested, we have classified each detection by recording in the table. This demonstrates that all recordings were reliably detected by all the devices at a close range and there is no single recording that is systematically under recorded compared to the others. See Line 306 for the amended table.

Line 315: Latter

Response: Amended as suggested

Line 315: Remove Parenthesis

Response: Amended as suggested

Line 316: ?

Response: It looks like the line number ended up in the table, not sure how that happened but it seems to have resolved itself in the revised manuscript.

Line 320: why not for the f-pod? was this in the discussion or i missed it?

Response: This factor was not retained in the F-POD model. Implication of this added to discussion, see comment below.

Line 324: I'm missing the AIC values for each model selection.. should include this in the supplementary

Response: Added AIC tables to supplementary -S2 File.

Line 324: affecting

Response: Amended as suggested

Line 324: do these dashes mean that the factor was excluded in the model selection? state this in the caption. what does it imply that the source level is not relevant for the final model for f-pod?

Response: Amended as suggested in caption. Sentence regarding the potential implication for the F-POD added to the discussion, see line 421.

Line 324: this is also significant

Response: Amended as suggested

Line 327: This is an important figure. Should be included in the paper and not as a supplementary. And better to combine all 3 graphs into one.

Response: We tried combining all 3 graphs into one and it was difficult to differentiate between the devices due to the overlapping 95% CI. As such, for clarity, we have kept the 3 separate graphs. This figure has been removed from supplementary and has been included in the paper. See Fig 2.

Graph amended. All three detection function graphs combined and added to main text

Line 333: “per device”

Response: Amended as suggested

Line 335: “Using”- “the”, “to represent"- “which represents

Response: Amended as suggested

Line 349: the word lower gives an impression that this is about vertical distance rather than horizontal. It is implied by EDR, but to make it easier to read I suggest using the word "smaller" or lesser"

Response: Amended as suggested

Line 349: insert EDR "the EDR for"

Response: Amended as suggested

Line 356: “device” not devices

Response: Amended as suggested

Line 371: emphasize not emphasizes, need not needed

Response: Amended as suggested

Line 373: Remove comma and replace with “.” Or separate into two sentences

Response: Amended as suggested

Line 375: Protection not protections

Response: Amended as suggested

Line 377: Important to mention the exact month/season of sampling for comparison

Response: Exact month has now been mentioned within the methods and results for comparison and context.

Line 390: “if” not when

Response: Amended as suggested

Line 392: “respectively” not respectfully

Response: Amended as suggested

Line 395-397: insert Table no.

Response: Amended as suggested.

Line 405-406: which study? has this only been reported for c-pods and not the other PAM devices?

Response: Reference added to Nuuttila et al.

Line 424: could be potentially

Response: Amended as suggested

Line 428: insert p-value, or table no.

Response: Amended as suggested.

Line 471: “sensitive species” is unclear

Response: Clarity added, amended to “species of concern for conservation and management”

Line 472: not just report, but also normalize/weight abundances/estimates using EDR/EDA

Response: This is a good point! Amended “If feasible, cetacean monitoring projects using static PAM devices should determine the EDR/EDA to aid comparability of detection rates and to normalise density and abundance estimates across sites and studies”

Figure 2: Figure can be improved, circles are not aligned.. important to mention the date/time of playback experiments used to calculate these values in the caption

Response: Thank you for pointing that out. Figure has been aligned properly and date of the experiment has been added to the figure caption.

Reviewer #2: Dear Authors

Thank you for conducting the research and preparing this manuscript comparing the detection performance of commonly used acoustic recorders. I found the manuscript to be well written, concise, and interesting. I think this manuscript will be of great interest to the audience of this journal. I don't have a lot comments except for the few in the marked up document.

Response: We thank the reviewer for taking the time to review this manuscript and for their positive reception. Please see responses to specific comments below.

Detailed comments:

Key words: Maybe replace these words on the title with words that are not on the title for easier indexing.

Response: Amended as suggested

Line 203: Shouldn't this value be negative? Hydrophone sensitivities are usually " negative because the amount of voltage produced by a sound is less than 1 volt per 1 pascal, the standard underwater pressure used for measurement".

Response: Yes, thanks for noticing this omission. Amended as suggested.

Line 275: Write number name for numbers below 10., Write out name in full.

Response: Amended as suggested

Line 378: Double of 0.153 is 0.306, so 0.279 is less than 0.306. I think what you can say here is "almost double".

Response: Amended as suggested

Line 430: What about changes in water temperature over time? Did you look at this?

Response: The experiment was conducted over a matter of hours therefore in relatively stable conditions, no changes in water temperature where found over this time period. However, as per comments of Reviewer 2, the date the experiment was conducted, as well as water temperature have been added to the manuscript, as water temperature is indeed an important factor that influences sound propagation, so it is important to have this context.

Line 443: “Two-fold higher”- I don't think the difference is two-fold based on Table 4, can you please recheck this?

Response: I checked Table 4 and this is correct. The difference between the EDR for C/FPOD was 31-33m, compared to 73m for the soundtrap. I’ve reworded the sentence slightly for clarity. “The EDR of buzzes was lower than that of clicks for all devices. However, the difference in EDR for the SoundTrap was twice as large as the difference observed for the other devices”

---

## [Decision Letter · Decision Letter 1]

14 Jan 2025

PONE-D-24-43959R1Can you hear me? Playback experiment highlights porpoise detection range differences between commonly used PAM devices: C-POD, F-POD, and SoundTrap.PLOS ONE

Dear Dr. Todd,

Thank you for submitting your manuscript to PLOS ONE. After careful consideration, we feel that it has merit but does not fully meet PLOS ONE’s publication criteria as it currently stands. Therefore, we invite you to submit a revised version of the manuscript that addresses the points raised during the review process.

We look forward to receiving your revised manuscript.

Kind regards,

Vitor Hugo Rodrigues Paiva, Ph.D.

Academic Editor

PLOS ONE

**Journal Requirements:**

Reviewers' comments:

Reviewer's Responses to Questions

**Comments to the Author**

1. If the authors have adequately addressed your comments raised in a previous round of review and you feel that this manuscript is now acceptable for publication, you may indicate that here to bypass the “Comments to the Author” section, enter your conflict of interest statement in the “Confidential to Editor” section, and submit your "Accept" recommendation.

Reviewer #2: All comments have been addressed

Reviewer #3: (No Response)

2. Is the manuscript technically sound, and do the data support the conclusions?

Reviewer #2: Yes

Reviewer #3: Partly

3. Has the statistical analysis been performed appropriately and rigorously? 

Reviewer #2: Yes

Reviewer #3: I Don't Know

4. Have the authors made all data underlying the findings in their manuscript fully available?

Reviewer #2: Yes

Reviewer #3: Yes

5. Is the manuscript presented in an intelligible fashion and written in standard English?

Reviewer #2: Yes

Reviewer #3: Yes

6. Review Comments to the Author

**Reviewer #2:**  Thank you for the revised manuscript. My comments and suggestions were properly addressed, and I have no further comments.

**Reviewer #3:**  General comments

I think this is a very useful experiment and paper but it could use more detail and clarification. I realize this is not the first set of review comments and in the detailed comments below I have made an effort to be explicit about what information would be helpful. An additional sentence or two in the identified paragraphs should provide the necessary information. If the results from one of the gain settings were excluded, this needs to be explained and the total number of playbacks included in the analysis should be consistent. If the results from both settings were used, I would suggest explaining why they were combined in the results.

The rationale behind and composition of the playback methods, especially where the method differs from those in the cited methods papers, should be justified. For example, Nuuttila et al. (2018) reported EDR results for both artificial and recorded harbour porpoise calls and by detected versus classified click trains (Figure 4). The selection of the five recordings is not explained. A previous reviewer questioned the duration of the recordings, and from the manuscript it is not clear how the 0.55 - 2.8 sec clips were combined into a one minute playback, how many times the clips were played, or at what gain. I understand a bit better after reviewing the playback2.2.csv file on your Github, but this information needs to be in the manuscript. A diagram of how the playbacks were composed would help the reader and even a condensed version of the results spreadsheet could be included in an appendix.

I would like more information to understand how your analysis supports the comparison of EDR for the recorded buzzes versus clicks. My understanding is that you have calculated the source level of the playback but are assuming the difference in these values could represent the difference in true source levels. However, this seems like a coincidence as in theory the buzz could have been detected at a higher received level than the click train (in the pool) depending on the distance and orientation of the captive harbour porpoise from the receiver. You reference the 8dB difference in the discussion, but I would recommend briefly explaining your rationale and assumptions in the methods. The other differences between the recordings (duration, number of clicks, frequency) and how this might impact detection range should also be discussed.

It should be clear throughout the manuscript that you have estimated detected range of the playbacks and not of harbour porpoise, given the source levels of the playbacks are much lower than the maximum source levels referenced. The title of the manuscript might imply that the values reported represent harbour porpoise detection ranges, but the emphasis should be on comparing the devices.

I would suggest you review the use of the word “their” (when referencing the instruments) and the term “vocalisations” throughout the manuscript. I realize this is a common practice, but as clicks are not produced using vocal cords, some odontocete scientists prefer to use “clicks”, “acoustic signals” or “calls”.

Detailed comments

Line 54-55: beaked whale researchers are transitioning from “Cuvier’s” to “goose-beaked”

Line 65: change “their successor” to “its successor”

Lines 92-96: Does the literature suggest why FPODs and SoundTraps detected more than C-PODs other than ambient noise? Here or in the methods would be a good place to provide a bit more information about how the click train detectors work on C-PODs and F-PODs as this likely has a significant impact on what would be detected in a long term monitoring study.

Line 97: change “...their detection ranges…” to “its detection range” or “the detection range”

Line 100-101: would suggest a less broad statement such as “assessing the effect of porpoise call type including echolocation clicks and foraging buzzes” and I still recommend that more information is needed about why the recordings selected can be representative of these two types of echolocation.

Line 114: What frequency was the high pass filter set to?

Line 131-148: An explanation of how / why the five recordings were selected would be helpful, and why artificial signals were not used. You could also explain how / why Rec3 and Rec5 are representative of the two echolocation types.

Line 140-142: Here general observation clicks are defined as being regular echolocation clicks but in Table 1 these recordings include buzzes. I would suggest updating the paragraph or the table for consistency.

Line 144-145: Does off axis echolocation refer to a signal being emitted off centre or the recording of a signal outside of the beam? This sentence could be improved with some rephrasing.

Line 155: I assume dBpp refers to peak to peak but it is helpful to state that clearly when it is first referenced as sometimes dBpp is used to denote zero to peak.

Lines: 154-159: I would recommend briefly addressing these points again in your discussion. (1) The source levels of the playbacks were much lower than maximum porpoise source levels. A brief discussion of how this does or does not impact your results would be helpful. (2) The source levels of the captive porpoise was not calculated. How is the source level of the playback relevant to the comparison of the different click types? (2) Click characteristics from these recordings might be different than those in the wild. Does this impact your results and conclusions? Is there any information about how C-POD and F-POD detectors perform with captive animal signals? It would be helpful to see that you have considered these factors in your interpretations and conclusions.

Line 164-165: It would be helpful if you explained how you decided on the bearings of the two transects.

Line 186: I would suggest replacing “circa” with “approximately”

Line 188-190: This should be reworded for clarity. It is not clear how the playbacks were composed and it is difficult to follow the numbers referenced here and in the results. In addition, if each recording (n = 5) was played at two gain settings, the use of the gain and how it impacts the results should be clarified. Why were those gain settings selected? Line 190 states there were 10 playbacks per station but in the results (Table 2) it is reported that there were 8 playbacks per recording and 40 playbacks at each station. Was each of the 5 recordings played 4 times at each station (given there were two replicates)? Was the sequence of recordings (Rec1, Rec2, Rec3, Rec4, Rec5, Rec1, Rec2, … etc.) repeated 4 times within one minute? What was the buffer (time gap) between the recordings? How were the two gain values used? From the playback2.2.csv file on your Github it seems that a recording was repeated a certain number of times to make up one minute, and this repeated sequence was played sequentially. This needs to be explained in your methods. It is difficult to evaluate the results from the CPOD or FPOD click detector or the manual review of the sound data in Audacity without understanding how these recordings were compiled and how the playbacks were conducted. A diagram or spectrogram of how the recordings were composed - i.e. what was played at each station - would be helpful.

Line 205-208: Given signals were transmitted from a transducer using two different gain settings, I would expect a mean source level or two different estimates of source level. In the playback2.2.csv file on your Github, source levels for playbacks at the 50 dB gain settings are reported. If you excluded the results from one of the gain settings, this needs to be explained - there are 240 rows in your results spreadsheet, if half were excluded the total number of playbacks should add up to 120 (your results report 240 playbacks)? Also, please explain why one SPL measurement was taken across the full spectrum, as often the focus should be on the peak frequencies of interest. Please clarify how you measured and calculated SPL and SL from one or more playbacks of the five different recordings.

Line 208: Needs a period

Line 214-215: Can you provide more detail about how you defined a “valid detection”. Nuuttila et al. (2018) provide details that would be helpful here, e.g., they recorded the number of clicks detected and whether only part of the sequence was recorded. They also divided C-POD results based on whether clicks were detected in the raw files, by the click train detector, or as porpoise click trains. Use of results from only raw files should be explained as this is relevant to how well a C-POD or F-POD will perform in long term studies - when the click detector and classification as NBHF will be relied on.

Line 221-225: Similar to my questions above, how was a detection validated? Did the entire one minute recording have to be detected or did partial recordings count?

Line 234: Was the replicate or gain used on the transducer not relevant?

Line 261: Should “wheras” be “wherein” or “within which”?

Line 263: Missing a “)”

Line 284-288: The detection of these two recordings, especially on the F-POD and C-POD, could differ due to factors such as frequency, duration, number of clicks, and amplitude. A justification for why those recordings were selected as representative of two echolocation behaviours would be helpful. In addition, if the assumption is that the difference in playback source level is representative of the difference in source level of an echolocation click train versus buzz click train, this needs to be explained and justified by literature.

Line 295: This section and the title of Table 2 contains some information that would be helpful in the methods when explaining how the playbacks were designed and conducted. In the methods (Line 188) states that each playback is the transmission of each recording per gain setting. There is no reference to the different gain settings in the results. This should be clarified in the methods and results. The playback2.2.csv file on your Github shows that the engine as one for the start of your playbacks. Even if this does not impact results, it should be noted. Furthermore, for both tracks 1.1 and 1.2 the measured distance (“true distance” column) from the mooring for stations at 100 and 200 m are recorded as 128.75 m and 225.97 m. This should be explained. Did your include these true distances in your distance analysis or models? If not, you could explain why you assumed it would not alter your results.

Line 300: The F-POD detected 90/240 playbacks, 18 were marked as click trains, and two as NBHF. This represents fairly poor performance by the F-POD; I would expect due to the source level, short duration of the recordings and limited number of clicks. A brief discussion of this and some more details on how the click train detector and NBHF classifier works would be helpful - does it rely on a certain number of clicks to identify a click train? I think it is relevant to know which of the recordings were identified as click trains and which of those were identified as NBHF. This is also why it would be helpful to understand the rationale behind the recordings selected and experiment design. I suggest this is relevant because for long term monitoring I expect the click detector and NBHF classifier would be relied on, as opposed to manual review used to find the 90 detections.

Table 2: Including percentages would be helpful as it is difficult to determine how many playbacks were detected out of the total played for each recording and each station (distance). For example, 100 playbacks were detected by the SoundTrap, which is ~42%. If two gain settings were used, that information should also be in the table.

Line 327: I would suggest including a sentence that reports the EDV values before comparing them as percentages.

Line 434: Replace the “&” with “and”

Line 348: suggest replacing “therein” with another word as it’s not the correct use

Line 350: change “...their detection rates…” to “... detection rates…”

Line 354: change use of “their”

Line 359: suggest adding a period after C-POD and making this two sentences

Line 360: Is “precision” the correct word here?

Line 361-363: The two recordings used as representative of call types had different source levels, but this has not been linked to the source level of the captive porpoise. If you are assuming that the difference in source levels, as well as other characteristics that impact detection on the devices, is representative of the difference in clicks and buzzes, this needs to be explained. In addition, it should be clear that you have estimated detected range of the playbacks and not of harbour porpoise given the source levels of the playbacks are much lower than the maximum source levels referenced. Given you should have source level and received level values, you could calculate transmission loss and estimate detection range with the theoretical maximum source level.

Line 393: “subsequently” means “afterward”, I would suggest selecting a different word here

Line 400-401: Detection range of wild porpoise could also be greater than experimental estimates depending on environmental conditions, devices, and source levels. You reference this at line 381.

Line 406: Line 397 states that EDR has varied considerably between devices, was this due to a calibration issue?

Line 418: Does this reference the calculated source level of the different recordings or the change in gain used for the transducer?

Line 422: Wouldn’t amplitude of a signal be related to source level?

Line 425: I’m not sure the difference in bearing between the two transects provides enough information to make conclusions about how the orientation of the devices impacted detection. I assume the devices might rotate or turn while in the water column?

Line 442-445: I suggest a brief explanation for why the difference in source level of these two playbacks was assumed representative of the difference between click and buzz source level be provided in the methods. Other than the reduced source level, is it possible the frequency, duration, or number of clicks impacted whether these two recordings were detected? If not, why?

Line 453-455: Do you have a reference for this? Directionality of the echolocation beam impacts detection range but is it more an issue of detection probability?

7. PLOS authors have the option to publish the peer review history of their article (what does this mean? ). If published, this will include your full peer review and any attached files.

**Do you want your identity to be public for this peer review?** For information about this choice, including consent withdrawal, please see our Privacy Policy .

Reviewer #2: No

Reviewer #3: No

---

## [Author Response · Author response to Decision Letter 2]

25 Feb 2025

Thank you for thoroughly reviewing this manuscript; we greatly appreciate the time and effort. We have made significant improvements based on the comments provided. Please see the Response to Reviewers for detailed responses.

---

## [Editor Report · Decision Letter 2]

27 Feb 2025

Can you hear me? Playback experiment highlights detection range differences between commonly used PAM devices: C-POD, F-POD, and SoundTrap.

PONE-D-24-43959R2

Dear Dr. Todd,

We’re pleased to inform you that your manuscript has been judged scientifically suitable for publication and will be formally accepted for publication once it meets all outstanding technical requirements.

Kind regards,

Vitor Hugo Rodrigues Paiva, Ph.D.

Academic Editor

PLOS ONE
---

## [Editor Report · Acceptance letter]

PONE-D-24-43959R2

PLOS ONE

Dear Dr. Todd,

I'm pleased to inform you that your manuscript has been deemed suitable for publication in PLOS ONE. Congratulations! Your manuscript is now being handed over to our production team.

Kind regards,

on behalf of

Dr. Vitor Hugo Rodrigues Paiva

Academic Editor

PLOS ONE